# Analysis of New RGB Vegetation Indices for PHYVV and TMV Identification in Jalapeño Pepper (*Capsicum annuum*) Leaves Using CNNs-Based Model

**DOI:** 10.3390/plants10101977

**Published:** 2021-09-22

**Authors:** Arturo Yee-Rendon, Irineo Torres-Pacheco, Angelica Sarahy Trujillo-Lopez, Karen Paola Romero-Bringas, Jesus Roberto Millan-Almaraz

**Affiliations:** 1Facultad de Informática Culiacán, Universidad Autónoma de Sinaloa, Culiacán 80013, Mexico; arturo.yee@uas.edu.mx (A.Y.-R.); asarahy_96@hotmail.com (A.S.T.-L.); kpaolarb@gmail.com (K.P.R.-B.); 2Facultad de Ingeniería, Universidad Autónoma de Querétaro, Santiago de Querétaro 76010, Mexico; irineo.torres@uaq.mx; 3Facultad de Ciencias Físico Matemáticas, Universidad Autónoma de Sinaloa, Culiacán 80010, Mexico

**Keywords:** deep learning, vegetation index, plant viruses, transfer learning, data augmentation

## Abstract

Recently, deep-learning techniques have become the foundations for many breakthroughs in the automated identification of plant diseases. In the agricultural sector, many recent visual-computer approaches use deep-learning models. In this approach, a novel predictive analytics methodology to identify Tobacco Mosaic Virus (TMV) and Pepper Huasteco Yellow Vein Virus (PHYVV) visual symptoms on Jalapeño pepper (*Capsicum annuum* L.) leaves by using image-processing and deep-learning classification models is presented. The proposed image-processing approach is based on the utilization of Normalized Red-Blue Vegetation Index (NRBVI) and Normalized Green-Blue Vegetation Index (NGBVI) as new RGB-based vegetation indices, and its subsequent Jet pallet colored version NRBVI-Jet NGBVI-Jet as pre-processing algorithms. Furthermore, four standard pre-trained deep-learning architectures, Visual Geometry Group-16 (VGG-16), Xception, Inception v3, and MobileNet v2, were implemented for classification purposes. The objective of this methodology was to find the most accurate combination of vegetation index pre-processing algorithms and pre-trained deep- learning classification models. Transfer learning was applied to fine tune the pre-trained deep- learning models and data augmentation was also applied to prevent the models from overfitting. The performance of the models was evaluated using Top-1 accuracy, *precision*, *recall*, and *F1-score* using test data. The results showed that the best model was an Xception-based model that uses the NGBVI dataset. This model reached an average Top-1 test accuracy of 98.3%. A complete analysis of the different vegetation index representations using models based on deep-learning architectures is presented along with the study of the learning curves of these deep-learning models during the training phase.

## 1. Introduction

Agriculture is facing harder yield scenarios every year, and this is due to factors such as abiotic and biotic stress conditions. On the one hand, abiotic stress includes physical problems such as non-favorable environment conditions and chemical stress due to toxic conditions which may be present in soil, air, or irrigation water [1]. On the other hand, biotic stress conditions are divided into visible pests and microorganisms that lead to plant diseases [2]. Visible pests are mainly insects, but also rodents and any kind of animals which can result in harm to the crop [3]. However, microorganisms are the main concern regarding agricultural losses which account for around 20–40% of yearly crop yields worldwide [2].

Plant diseases are a major concern for agriculture where bacteria, fungi, and virus diseases are the main issues [2]. However, bacteria- and fungi-based diseases may be a problem, but they can be controlled by using specific commercial treatments [1]. On the contrary, virus-based diseases are difficult to control due to the lack of anti-viral treatment where vector interference is the most common strategy. Furthermore, there are many actions that can be performed to assist in the control of plant virus-based diseases such as early symptom detection and vector control [4].

Currently, precision agriculture is defined as the integration of a wide variety of information technologies to help to maximize agricultural yield and minimize costs and agricultural losses [5]. These information technologies include satellite imaging, sensor networks, image processing, computer vision, and artificial intelligence to name but a few [6]. Because of this, image processing is considered one of the most frequently utilized technologies for plant disease detection where the main objective is to identify the pathogen symptoms of the plant [7]. There are several methods and approaches to achieve this, where each combination of plant–pathogen may be a case study. The most common symptoms of plant diseases are chlorosis, necrosis, and spots, among others [8]. Therefore, each pathogen has its own visual damage patterns on plant tissues which may be a problem for image-processing and computer-vision algorithms by sometimes making them unsuitable in identifying a wide range of diseases using the same algorithm [9]. In addition, vegetation indices are image-processing algorithms focused on highlighting plant features such as lesions and many stress responses. These algorithms are based on linear algebra operations between image components such as red, green, and blue masks on visible light RGB images and specific multispectral bands images such as red edge (RE) and near infrared (NIR), among others [10].

Machine learning (ML) is a branch of Artificial Intelligence (AI) that provides learning techniques to build computer models that emulate the way that humans learn from data. ML is growing fast, and it is proving to be efficient in identifying between object classes such as healthy and infected plants [11]. ML is essentially divided into supervised learning, which is when a model is trained using labeled pre-defined training data, and unsupervised learning, which is when a model uses unlabeled training data and discovers patterns and relationships in that data. The traditional machine learning techniques are based on statistical procedures and previous knowledge of the data [12]. Deep learning is a subset of ML based on a set of learning techniques that attempts to model high-level abstractions in data using multiple processing layers [13]. Several investigations are currently using deep-learning techniques, mainly convolutional neuronal network (CNN) architectures for the study of plant–pathogen interactions such as VGG-16 [14] and inception [13], among others. However, previous investigations have applied deep-learning algorithms on raw RGB data without considering the previous image improvements of new vegetation indices.

Recently, deep-learning techniques have become the foundations for many breakthroughs in the automated classification of plant diseases using digital images. In the agricultural sector, many recent visual-computer approaches have used different deep-learning classification models. CNNs are used extensively as a powerful class of classification models of digital images in a variety of problems in the agricultural field, such as: plant species identification, plant disease identification, crop pest classification, among others.

Many state-of-the-art approaches for agricultural pest identification and classification use CNNs as classification models. In 2016, Liu et al. [15] proposed an eight-layer CNN network based on AlexNet to learn local features from the insect image dataset along with a saliency map-based approach for localizing pest insect objects in natural images. The authors reported a mean accuracy precision of 0.951. In the research of Wang et al. [16], the authors used two CNN architectures, LeNet-5 and AlexNet, to classify pest images. A total of 82 common pest types were classified, with an accuracy of 91%. In 2017, Cheng et al. [17] proposed a deep-residual-learning model to identify 10 classes of agricultural pests under complex farmland background. The accuracy of the deep-learning model reached 98.67%. Lu et al. [18] investigated rice disease identification using a deep CNN model to identify 10 common rice diseases. The authors used a dataset of 500 natural images of diseased and healthy rice leaves and stems captured from an experimental rice field. The proposed model achieved an accuracy of 95.48%. In 2018, Barbedo [19] studied the effects of using relatively small datasets on the effectiveness of deep-learning tools for plant disease classification. The experiments were carried out using an image database containing 12 plant species, each presenting very different characteristics in terms of number of samples, number of diseases, and variety of conditions. In 2019, Thenmozhi and Srinivasulu [20] proposed an efficient deep CNN model to classify insect species on three insect datasets. The National Bureau of Agricultural Insect Resources (NBAIR) dataset consists of 40 classes of field crop insect images, the Xie1 and the Xie2 datasets contain 24 and 40 classes of insects, respectively. The proposed model was evaluated and compared with pre-trained deep-learning models based on AlexNet, ResNet, GoogLeNet, and VGGNet for insect classification. Transfer learning was applied to fine tune the pre-trained models. Data augmentation techniques such as reflection, scaling, rotation, and translation were also applied to prevent models from overfitting. The highest classification accuracies of 96.75%, 97.47%, and 95.97% were achieved in the proposed CNN model for NBAIR, Xie1, and Xie2 insect datasets, respectively. Picon et al. [21] proposed a crop-conditional CNN architecture that seamlessly incorporates contextual meta-data consisting of the plant species identification. To validate this approach, the authors generated a challenging dataset that consists of more than 100 images taken by cell phone in real field conditions. This dataset contains almost equally distributed disease stages of 17 diseases and 5 crops (wheat, barley, corn, rice, and rapeseed). The proposed crop conditional CNN model obtained an average balanced accuracy of 0.98 and removed 71% of the classifier errors. In 2020, Barman et al. [22] presented a comparison of CNNs for smartphone image citrus leaf disease. The MobileNet CNN and Self-Structured CNN architectures were trained and tested on the same citrus dataset. The best training and validation accuracies obtained for Self-Structured CNN were 98% and 99% at epoch 12. Kang and Chen [23] developed a fast implementation framework for deep-learning-based fruit detection. This framework includes a label generation module and a fruit detector LedNet. The LedNet model with a lightweight backbone achieved 0.821 and 0.853 on recall and accuracy of the apple detection. Alves et al. [24] proposed a new deep-residual-learning model called ResNet34 to address cotton pest recognition issues using a field-based image database of 1600 images balanced for 15 pests and 1 with no insects. The classification model achieved a high accuracy with F1-score of 0.98.

The novelty of this project is the development of a new predictive analytics methodology to identify Tobacco Mosaic Virus (TMV) and Pepper Huasteco Yellow Vein Virus (PHYVV) visual symptoms on Jalapeño pepper (*Capsicum annuum*) leaves by means of new RGB vegetation indices as pre-processing algorithms and pre-trained deep-learning models based on VGG-16, Xception, Inception v3, and MobileNet v2 architectures. The objective of this methodology is to find the most accurate combination of an RGB vegetation index and a deep-learning classification model to identify PHYVV and TMV in Jalapeño pepper plants by using a small training dataset. Furthermore, it presents measurements of how identification accuracy can be improved by pre-processing images with new RGB vegetation indices and its usefulness for future research.

## 2. Results

All the experiments were carried out using Keras version 2.2.4, with Tensorflow 1.13.1 as backend, and Python version 3.7.8. Experiments were conducted on a PC with the following configuration: Intel Xeon W-2133 processors, 32 GB of RAM, and one NVIDIA GTX 1080 GPU.

### 2.1. Image Processing Results

The results obtained from the image processing are shown in Figure 1. It can be observed that image representations were organized as columns and dataset classes as rows. Consequently, an image results matrix is established to compare between different results where it is easy to observe that NRBVI and NGBVI indices are effective in highlighting chlorotic pixels in all the classes in their grayscale versions. It is also noteworthy that Jet colored versions of NRBVI and NGBVI are even more useful in identifying chlorotic areas inside the analyzed leaf. Furthermore, it is important to mention that comparing between rows, each class shows different patterns that need to be classified by an AI-based algorithm to finally have an efficient identification of infected plants.

### 2.2. Data Description and CNN Architectures Parameters

In this investigation, the experiments were conducted on five different datasets. One dataset consisted of the RGB images, whereas the other four were built applying each of the pre-processing algorithms previously described, that is, NRBVI, NRBVI-Jet, NGBVI, and NGBVI-Jet. Within each dataset there are three classes. These classes are labeled as healthy (non-infected leaves), PHYVV (PHYVV-infected leaves), and TMV (TMV-infected leaves). Each dataset has 100 images for each class, for a total of 300 images. Each dataset was divided into 80% (240 images) for training data and 20% (60 images, 20 images per class) for testing data. During the training phase, 20% of the validation data was randomly selected from the 240 images. These validation data are used to manually adjust hyper-parameters, which are essentially the settings that cannot be automatically learned during training. These include the learning rate and the batch size, among others.

The models were trained using the Stochastic Gradient Descent (SGD) as an optimizer with 0.9 momentum, and the learning rate was varied between 1e-3 and 1e-4. The learning rate defines the learning progress of the proposed model and updates the weight parameters to reduce the loss function of the network. The maximum number of epochs was set to 100 and batch sizes of 16 were used in this experiment. Table 1 provides a summary of the parameters and hyper-parameters used for each architecture.

### 2.3. Experimental Environment

Two strategies were followed for training the models used in these experimental procedures. First, the models were trained from scratch. That is, all trainable parameters in the model start with a randomly assigned value, and the second value is transferred from learning. That is, all trainable parameters start with a value that was assigned while training the model on a different dataset, in this case, transfer learning from pre-trained models on ImageNet dataset was used.

As opposed to with the traditional machine learning techniques, CNNs-based models can learn the features needed to discriminate between the classes directly from the original images instead of extracting the specific features manually. It is well known that training deep CNNs-based models requires many images. In this approach, the dataset is limited to about 300 images for each vegetation index representation (in each individual dataset). To address this issue, data augmentation was used to obtain a variety of conditions in the dataset. The input images were randomly rotated between 0 and 360 degrees, randomly translated along the X or Y direction (or both directions), and randomly flipped vertically or horizontally during training.

The training procedure was as follows. For each dataset and CNN architecture, a total of 10 trials were carried out for each model from scratch and 10 trials were carried out from transfer learning, with a random selection of 20% for validation data.

To evaluate the performance of the different models, the following metrics were computed:The average, maximum, and minimum Top-1 accuracies for each model. Top-1 accuracy measures the number of times the answer with the highest probability given by the models matches the expected answer. It is presented as the ratio of the number of correct answers to the total number of answers.The *precision* of each class. This is the ratio of the number of correctly predicted positive instances to the total number of predicted positive instances, see Equation (1). An associated term, macro-precision, is also used and it measures the average *precision* per class.
(1)$$precision=\frac{TP}{TP+FP}$$
3.The recall of each class that is the ratio of the number of correctly predicted positive instances to the total number of instances in a class, see Equation (2). Macro-recall measures the average recall per class.
(2)$$recall=\frac{TP}{TP+TN}$$
4.*F*1*-score* is the weighted average of precision and recall, see Equation (3). Macro-*F1-score* measures the average *F*1*-score* per class.
(3)$$F1-score=\frac{2\times precision\times recall\;}{precision+recall}$$where *TP* (true positive), *FP* (false positive), *TN* (true negative), and *FN* (false negative) are technical terms for binary classifiers. Specifically, *TP* is the positive samples correctly classified, *FP* is the positive samples misclassified, *TN* is the negative samples correctly classified, and *FN* represents the negative samples misclassified.

#### 2.3.1. Models from Scratch

In this first experimental configuration. A total of 10 models were built from scratch for each dataset and CNN architecture. The results of the test data indicate a slightly better performance from models that used the NRBVI-Jet dataset compared to those that used other datasets. For the VGG-16-based model, the average Top-1 accuracy was 73.3%, for the Xception-based model it was 83.3%, for the Inception-based model it was 82.3%, and for the MobileNet-based model it was 77.5%. Table 2 summarizes the performance of the CNNs-based models.

Based on previous results, it was noted that all models based on VGG-16 and MobileNet reached similar average Top-1 accuracies. The models based on Xception and Inception that used the NRBVI-Jet dataset have a slightly better performance than others that used different datasets; the difference in models performance ranged from 0.4% to 8.3%.

Table 3 reports metric values of *precision*, *recall*, and *F1-score* from models that used the NRBVI-Jet dataset. As can be seen in Table 3, the Xception-based model reached better metrics values than the other models. It obtained 0.83 on Macro-*precision*, Macro-*recall*, and Macro-*F1-score*.

For the HEALTHY class, the VGG-16-based model obtained 0.74, 0.85, and 0.79 on *precision*, *recall*, and *F*1*-score*, respectively. The Xception-based model achieved 0.87, 1.0, and 0.93 on *precision*, *recall*, and *F*1*-score*, respectively. The Inception-based model obtained 0.8, 1.0, and 0.89 on *precision*, *recall*, and *F*1*-score*, respectively. The MobileNet-based model achieved 0.7, 0.95, and 0.81 on *precision*, *recall*, and *F*1*-score*, respectively.

For the PHYVV class, the VGG-16-based model obtained 0.79, 0.55, and 0.65 on *precision*, *recall*, and *F*1*-score*, respectively. The Xception-based model achieved 0.78, 0.7, and 0.74 on *precision*, *recall*, and *F*1*-score*, respectively. The Inception-based model obtained 0.72, 0.65, and 0.55 on *precision*, *recall*, and *F*1*-score*, respectively. The MobileNet-based model achieved 0.55, 0.55, and 0.55 on *precision*, *recall*, and *F*1*-score*, respectively.

For the TMV class, the VGG-16-based model obtained 0.7, 0.8, and 0.74 on *precision*, *recall*, and *F*1*-score*, respectively. The Xception-based model achieved 0.84, 0.8, and 0.82 on *precision*, *recall*, and *F*1*-score*, respectively. The Inception-based model obtained 0.82, 0.7, and 0.76 on *precision*, *recall*, and *F*1*-score*, respectively. The MobileNet-based model achieved 0.92, 0.6, and 0.73 on *precision*, *recall*, and *F*1*-score*, respectively.

Analysis of the accuracy during the training phase provides additional insight into how a particular model is performed. Figure 2 shows the learning curves of models that use the NRBVI-Jet dataset. For all models, an overfitting problem is clearly seen at early epochs. Moreover, the difference in the learning curves of the training and validation data was huge; it ranged from 12% to 20%. Therefore, the models reached overfitting.

#### 2.3.2. Pre-Trained Models and Data Augmentation

In this experimental configuration, transfer learning from pre-trained models on the ImageNet dataset was employed. Based on the experimental and empirical evidence, it was found that a complete fine-tuning technique instead of a feature extractor technique was the best option. It starts with a pre-trained model as a starting point and then completely re-trains it on the target dataset. Data Augmentation was also used to prevent models from overfitting.

Table 4 shows the test accuracy of the fine-tuned models on the different datasets. The best performance was obtained by the models that used the NGBVI dataset, an average Top-1 accuracy of 96.6% was achieved by the VGG-16-based model, an average Top-1 accuracy of 98.3% was obtained by the Xception-based model, and average Top-1 accuracies of 95% and 92.3% were obtained by Inception-based model and MobileNet-based model, respectively. The difference in model performance between the maximum and minimum values of the average Top-1 test accuracy was approximately 6.7%.

Table 5 details the performance metric values of *precision*, *recall*, and *F1-score* from models that used the NGBVI dataset for each class. For the HEALTHY class, all models obtained 1.0 on *precision*, *recall*, and *F*1*-score*.

For PHYVV class, the VGG-16-based model obtained 0.91, 1.0, and 0.95 on *precision*, *recall*, and *F*1*-score*, respectively. The Xception-based model achieved 0.95, 1.0, and 0.98 on *precision*, *recall*, and *F*1*-score*, respectively. The Inception-based model obtained 0.87, 1.0, and 0.93 on *precision*, *recall*, and *F*1*-score*, respectively. The MobileNet-based model achieved 0.9, 0.95, and 0.93 on *precision*, *recall*, and *F*1*-score*, respectively.

For TMV class, the VGG-16-based model obtained 1.0, 0.9, and 0.95 on *precision*, *recall*, and *F*1*-score*, respectively. The Xception-based model achieved 1.0, 0.95, and 0.97 on *precision*, *recall*, and *F*1-*score*, respectively. The Inception-based model obtained 1.0, 0.85, and 0.92 on *precision*, *recall*, and *F*1*-score*, respectively. The MobileNet-based model achieved 0.95, 0.9, and 0.92 on *precision*, *recall*, and *F*1*-score*, respectively.

Figure 3 shows the accuracy and loss curves obtained from models that used the NGBVI dataset on the training and testing data during the training progress. It is clearly seen that the models based on Xception and Inception have a great performance improvement; as they have lower losses and higher accuracies.

## 3. Discussion

Plant viruses are a major threat to sustainable and productive agriculture, causing significant economic losses worldwide. Several studies on automated plant virus detection have been conducted using machine-learning techniques. One of the major machine- learning techniques is the convolutional neural network, a type of deep-learning neural network that has become a very successful automated classification technique for image-based plant virus classification.

This study demonstrated that the proposed methodology is a powerful method for high-accuracy, automated PHYVV and TMV classification in Jalapeño pepper leaf images. This method avoids the complex and labor-intensive step of feature extraction (hand-crafted feature extractors) from images employing CNNs-based models. Furthermore, using different RGB image vegetation indices enabled investigators to obtain more accurate CNNs-based classification models.

Data representation plays a crucial role in the performance of CNNs-based models. In the first experimental configuration, the results of the test set showed that the most accurate model was the Xception-based model that used the NRBVI-Jet dataset. It reached an average Top-1 accuracy of 83.3%. The worst models were those that used NRBVI and NGBVI datasets. In the second experimental configuration, transfer learning from pre-trained models and data augmentation were used. The best average Top-1 accuracy was obtained by an Xception-based model that used the NGBVI dataset, with an average Top-1 accuracy of 98.3%. In this experimental configuration, all models significantly increased the average Top-1 test accuracies between 12–26% concerning the first experimental configuration of the models built from scratch. Therefore, it was concluded that transfer learning from pre-trained models and data augmentation helped CNNs-based models to reduce the overfitting and to reach high test accuracies, as shown in Table 4 and Figure 3; the difference between the accuracy curves of the training and the validation data are small in each trained model. This study therefore offers a promising avenue for virus classification using different RGB image vegetation indices along with CNNs-based models in relatively small image datasets. The source code of this article is publicly released and can be downloaded from https://github.com/jrmillan1983/PHYVV_TMV_CNN (Appendix A).

## 4. Materials and Methods

### 4.1. Field Site

The experiment was carried out in two different small-sized greenhouses located in Queretaro, Mexico at 20°42′20.39″ N 100°15′33.81″ W for site A and 20°42′23.4″ N 100°15′42.2″ W for site B and their exact location can be observed as a red circle within Figure 4. The experiment at site A was carried out from February 2020 to June 2020 and site B was utilized as experiment site from July 2020 to October 2020. Both greenhouses are in a semi-arid region at an elevation of 1800 m with an average temperature of 25 °C with below-freezing winter temperatures. The plant substrate was made as a mixture of clay 55, sandy 25, silt 20, organic matter 1.81 and DAP 1.4 (g/cm^3^). Furthermore, the irrigation system was a standard drop-based irrigation system with average agronomic practices for Jalapeño plants.

### 4.2. Biological Materials

The plant material used was *Capsicum annuum* L. Type Jalapeño cv. Don Pancho from the National Institute of Agricultural and Livestock Forestry Research (INIFAP, in Spanish), with a purity of 98%. The plants were grown in two small greenhouses with an average temperature of 25 °C which can be observed in Figure 5. The PHYVV sample was provided by Cinvestav, Irapuato, Mexico and the TMV samples were provided by UNAM-Iztacala, Mexico. In order to infect the plants, a bio-ballistic-based procedure was carried out to inoculate PHYVV into the plants [25]. With respect to TMV, plants of 6 to 8 leaves were abraded with wet carborundum (400 grit) and inoculated with 30 µg of TMV (25 µg/mL), by gently rubbing adaxial leaves’ surfaces. Furthermore, RT-PCR confirmation tests were carried out to ensure the presence of the virus inside the plants according to the procedure described by Guevara-Olvera et al. [26].

### 4.3. Image Dataset

The image dataset for this project was generated by taking photographs of Jalapeño pepper leaves using a RGB camera on a smartphone model Samsung Galaxy S7 edge at a 4032 × 3024 12 MP resolution with 4:3 aspect ratio at ambient light conditions, using no flash and a thick white paper background. It is noteworthy that to avoid being invasive the photographs were taken in situ without cutting the leaves. Furthermore, a dataset of plant leaves was generated by including 3 classes (healthy, PHYVV, and TMV) with 100 images within each class for a total of 300 images for training purposes. Sample images of these 3 classes can be seen in Figure 6.

### 4.4. Vegetation Index Algorithms

Vegetation indices are algorithms that are focused on estimating the fractional green vegetation cover by means of indirect estimation of vegetation surface through linear algebra operations between image masks such as red, green, and blue, or specific hyper-spectral bands [27]. Usually, vegetation indices are focused on quantifying vegetation covered areas for remote sensing applications. However, the most common plant disease symptoms are chlorosis and necrosis. Because of this, two new vegetation indices are proposed to highlight chlorosis and necrosis areas across vegetation with the objective of obtaining a better identification of chlorotic and necrotic symptoms.

#### 4.4.1. Normalized Red–Blue Vegetation Index

NRBVI is a newly proposed vegetation index focused on highlighting chlorosis areas and discarding non-vegetation pixels from the greenhouse environment such as walls and ground to name but a few. Furthermore, NRBVI can be estimated from RGB images by subtracting red from blue masks as the first step, followed by a further normalization operation by dividing the resulting pixels by the maximum pixel value as can be observed at Equation (4).
(4)$$\mathrm{NRBVI}=\frac{\mathrm{RED}-\mathrm{BLUE}}{\max\left(\mathrm{RED}-\mathrm{BLUE}\right)}$$

#### 4.4.2. Normalized Green–Blue Vegetation Index

Alternatively, NGBVI is another newly proposed vegetation index, which is focused on highlighting chlorosis symptoms too, but by means of green and blue masks. As can be observed in Equation (5), NGBVI can be calculated by subtracting green from blue masks in the numerator and further dividing it by the highest pixel value from the numerator result.
(5)$$\mathrm{NGBVI}=\frac{\mathrm{GREEN}-\mathrm{BLUE}}{\max\left(\mathrm{GREEN}-\mathrm{BLUE}\right)}$$

#### 4.4.3. Jet Color Scale

Color scales are a useful tool for highlighting small differences in grayscale images. Most commercial equipment uses color scales to improve visualization for humans in applications such as vegetation indices, thermal cameras, and different kinds of remote sensing applications. Color scales consist of a discretization of the 8-bit grayscale into sixteen solid color bands where colors are assigned depending on the color scale that you chose. For this project, the jet color pallet was chosen due to its popularity among vegetation index applications and for its clear visualization where blue corresponds to low intensity pixels and red to the highest values. For clarity, grayscale and jet scale differences are shown in Figure 7.

### 4.5. Deep-Learning Models

CNNs have become the most popular learning algorithms in vision-related applications in recent years. Some of the remarkable application areas of CNNs include image segmentation, classification, plant species identification, plant disease identification, crop pest classification, among others. There are a lot of different state-of-the-art CNN architectures, therefore testing of all the possibilities was not a viable option. In this article, four common deep-learning network architectures were used (VGG-16, Xception, Inception v3, and MobileNet v2).

One of the most popular CNN architectures is VGG. This algorithm represents one of the most successful state-of-the-art CNN architectures due to its powerful and accurate classification ability. VGG network was introduced by Simonyan and Zisserman in 2014 [28]. In this study, a VGG-16 architecture was used that consists of 16 sequentially stacked convolution layers followed by rectified linear units (ReLU). A MaxPooling is computed after every two convolutional layers in the first four layers, and then after every three layers in the rest of the network, ending with three fully connected layers followed by a softmax layer. The block diagram of a VGG-16 network is illustrated in Figure 8.

The Xception architecture is a linear stack of depthwise separable convolution layers with residual connections [29]. The Xception network consists of 36 convolutional layers that perform as the feature extractor. These 36 convolutional layers are organized into 14 modules with linear residual connections around them, except for the first and last modules, and are followed by a softmax layer. A complete description of the specifications of the network is featured in Figure 9.

Inception v3 is a convolutional neural network architecture from the Inception family. The Inception v3 network is composed of 11 Inception modules of five kinds in total and supports the following concepts.

Factorizing convolution to reduce the number of parameters (connections) without decreasing the network efficiency.Factorizing into smaller convolution to reduce the number of parameters. For example, a 5 × 5 convolution filter has 25 parameters, this filter is replaced by two 3 × 3 convolution filters, 3 × 3 + 3 × 3 = 18 parameters, this is a reduction of 28% of the number of parameters.Factorizing into asymmetric convolution is to factorize a standard two-dimensional convolution kernel into two one-dimension convolution kernels. For example, a 3 × 3 convolution filter (9 parameters) could be replaced by a 1 × 3 convolution filter followed by a 3 × 1 convolution filter (1 × 3 + 3× 1 = 6 parameters). This is a reduction of 33% on the number of parameters.Auxiliary classifier is a small CNN inserted between layers during training, and the loss incurred is added to the main network loss.Efficient Grid Size Reduction.

A complete description of the specifications of the network is featured in Figure 10. For further description of Inception v3, see Szegedy et al. [30].

MobileNet is a lightweight CNN architecture. The first version of MobileNet was proposed by Google in 2017 [32] and it is oriented to mobile and embedded devices. It is characterized by using depthwise and pointwise convolution. MobileNet v2 [33] adapts inverted residual blocks which can incorporate low-level features with high-level features. The architecture of MobileNet v2 contains the initial full convolution layer with 32 filters, followed by 19 residual bottleneck layers, see Table 6. For further description of MobileNet v2, see Sandler et al. [33].

### 4.6. Image Processing Experiment

The image-processing experiment setup consisted of applying the RGB image as common input to four different algorithms, which are NRBVI and NGBVI in grayscale as well as Jet colored versions, and the original RGB image, to compare the performance of each algorithm in order to identify chlorosis lesions on Jalapeño leaf images [34]. The image-processing methodology is described as block diagram in Figure 11, where the RGB image as common input and five versions of the same image as outputs of the whole process can be observed.

## 5. Conclusions

CNNs are used extensively as a powerful class of classification models of digital images in a variety of applications in the agricultural field, such as plant species identification, plant disease identification and crop pest classification, among others. This article proposes an innovative methodology for the identification of Tobacco Mosaic Virus (TMV) and Pepper Huasteco Yellow Vein Virus (PHYVV) visual symptoms on Jalapeño pepper (*Capsicum annuum*) leaves.

The proposed methodology combines new RGB vegetation indices and deep-learning classification models. The results show that the proposed methodology can correctly and effectively recognize TMV and PHYVV through image recognition of different image representations such as RGB, NRBVI, NRBVI-Jet, NGBVI, and NGBVI-Jet. The best CNNs-based models were those that used NGBVI vegetation index representations. The pre-trained Xception-based model reached an average Top-1 accuracy of 98.3%. Future investigations will focus on increasing the dataset size and analyzing other state-of-the-art CNN architectures.

## Figures and Tables

**Figure 1 plants-10-01977-f001:**
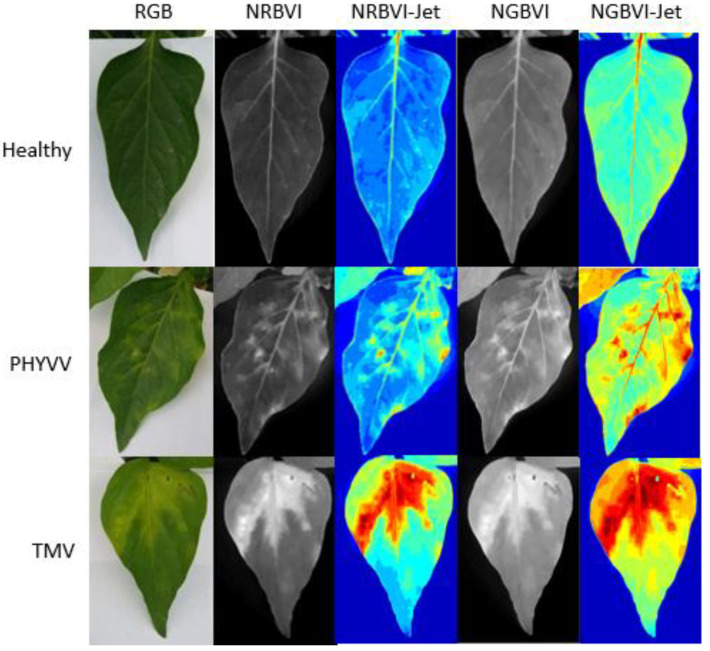
Image processing results. Algorithms are presented as columns and classes as rows.

**Figure 2 plants-10-01977-f002:**
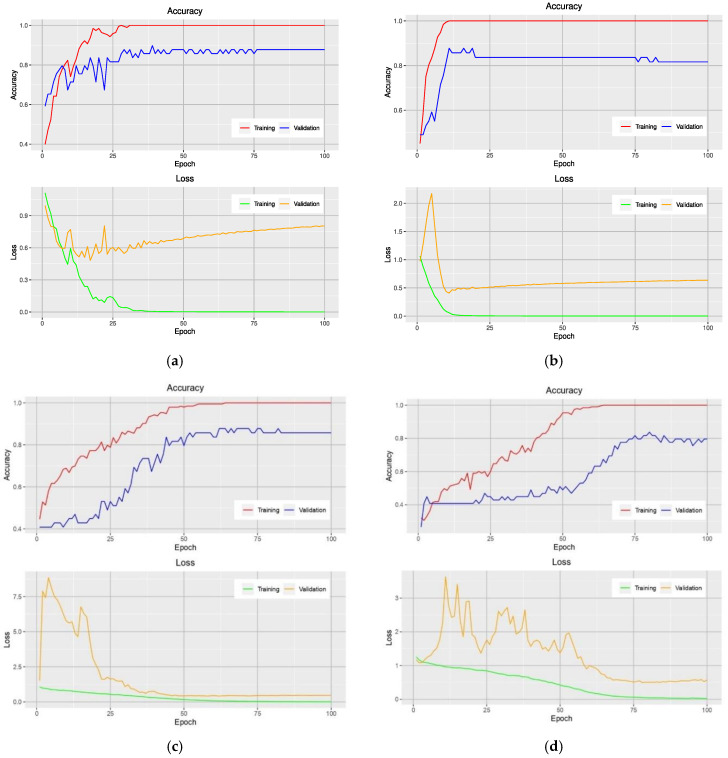
Learning curves of VGG-16-based model (**a**), Xception-based model (**b**), Inception-based model (**c**), and MobileNet-based model (**d**) on the training and validation data using the NRBVI-Jet dataset.

**Figure 3 plants-10-01977-f003:**
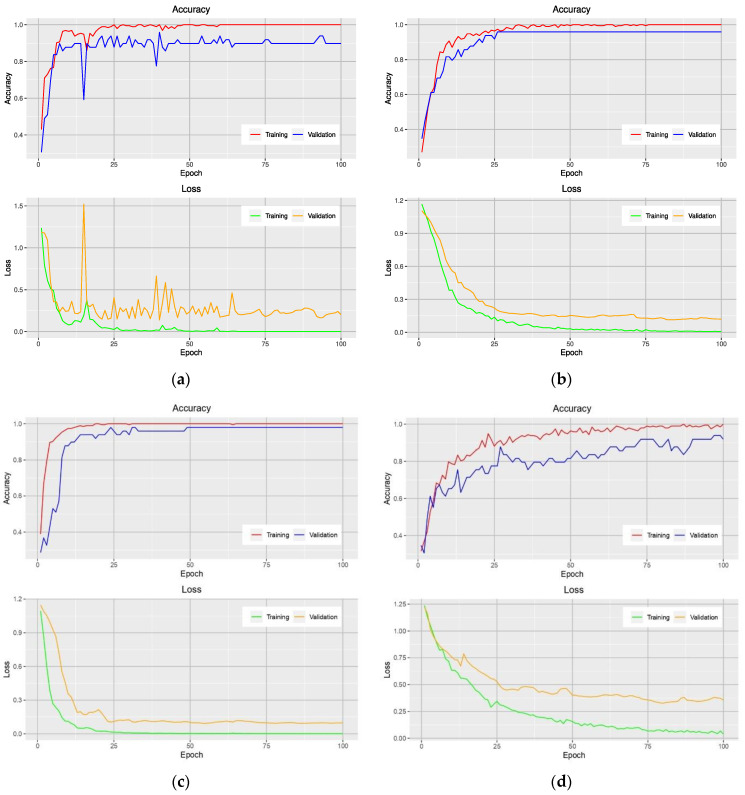
Learning curves of VGG-16-based model (**a**), Xception-based model (**b**), Inception-based model (**c**), and MobileNet-based model (**d**) on the training and validation data using the NGBVI dataset.

**Figure 4 plants-10-01977-f004:**
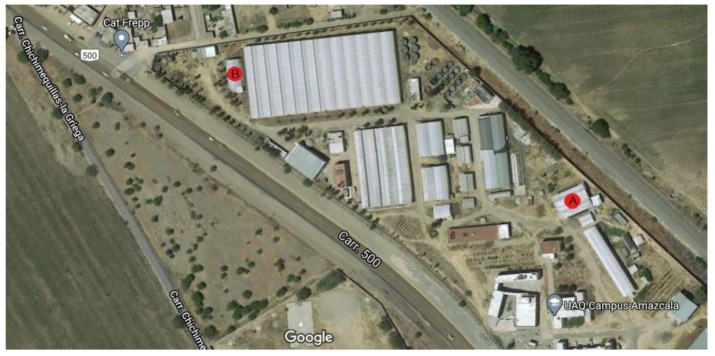
Experiment location in the field.

**Figure 5 plants-10-01977-f005:**
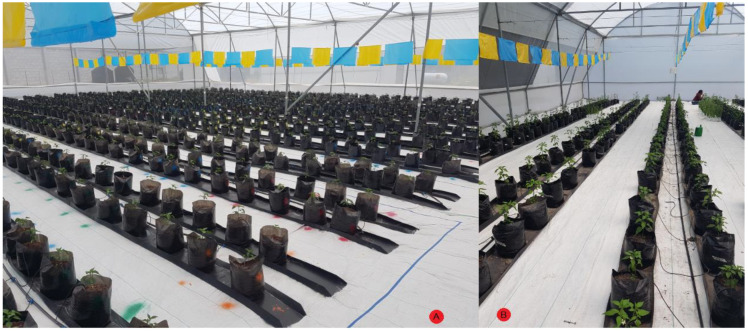
Experimental greenhouse with *Capsicum annuum* L. Type Jalapeño cv. Don Pancho. Site (**A**) left side and site (**B**) right side.

**Figure 6 plants-10-01977-f006:**
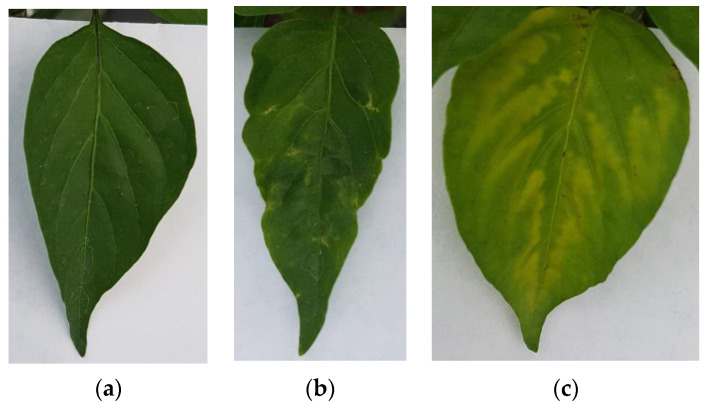
RGB image examples of the generated Jalapeño plant dataset. (**a**) healthy leaf, (**b**) PHYVV infected leaf and (**c**) TMV infected leaf.

**Figure 7 plants-10-01977-f007:**
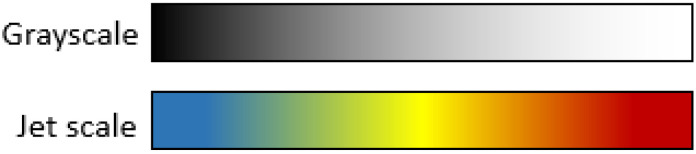
Grayscale and Jet scale equivalence.

**Figure 8 plants-10-01977-f008:**
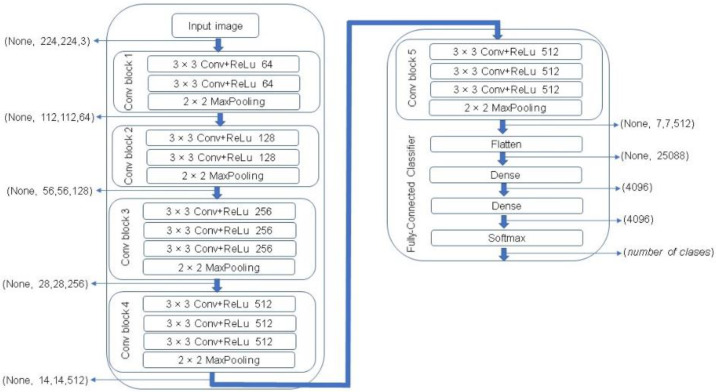
The Visual Geometry Group (VGG)-16 network architecture [28].

**Figure 9 plants-10-01977-f009:**
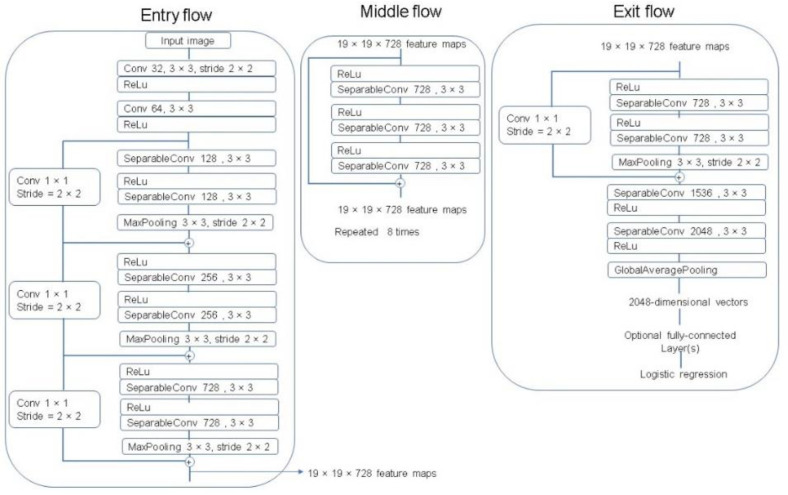
The Xception architecture [29].

**Figure 10 plants-10-01977-f010:**
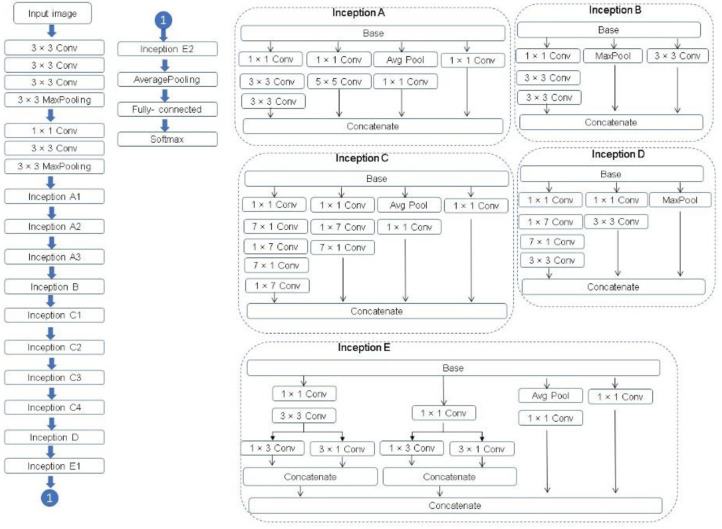
The Inception v3 architecture [30,31].

**Figure 11 plants-10-01977-f011:**
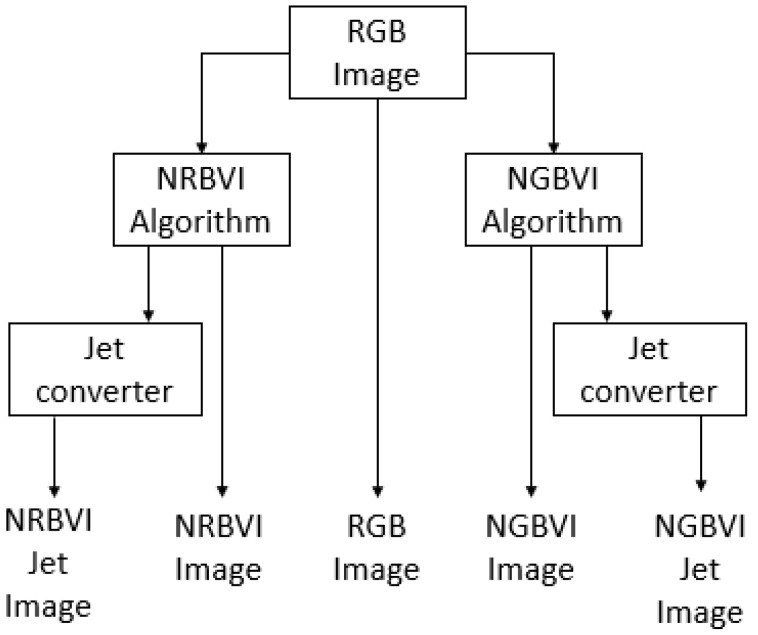
Block diagram for the image-processing experiment setup.

**Table 1 plants-10-01977-t001:** Parameter setting of CNN architecture models.

Parameters	VGG-16	Xception	Inception v3	MobileNet v2
Optimizer	SGD	SGD	SGD	SGD
Learning rate	1e-3	1e-4	1e-3/1e-4	1e-3/1e-4
Momentum	0.9	0.9	0.9	0.9
Weight decay	1e-4	1e-5	1e-5	1e-5
Epochs	100	100	100	100
Batch size	16	16	16	16

**Table 2 plants-10-01977-t002:** Recognition test accuracy of different models that were trained from scratch for different datasets (best results are highlighted with bold font).

Dataset	VGG-16 (avg/min/max)	Xception (avg/min/max)	Inception v3 (avg/min/max)	MobileNet v2 (avg/min/max)
RGB	73.3/71.4/75	80.3/79.5/81.6	77.5/71.4/79.5	73.4/69.3/75.0
NRBVI	71.6/70.1/80.2	75.0/74.4/81.6	76.6/75.5/77.5	70.6/65.1/71.6
NRBVI-Jet	**73.3/71.3/75.7**	**83.3/81.6/83.6**	**82.3/79.9/85.7**	**77.5/70.0/81.6**
NGBVI	70.0/59.1/70.4	76.6/63.3/76.9	75.5/68.3/79.5	70.0/66.6/73.4
NGBVI-Jet	73.1/65.3/75.2	82.9/80.2/85.7	81.6/79.5/83.6	71.4/70.0/73.4

**Table 3 plants-10-01977-t003:** Classification report of *precision*, *recall*, and *F*1*-score* of the models that use NRBVI-Jet dataset.

Class Name	VGG-16			Xception			Inception v3			MobileNet v2		
	*Precision*	*Recall*	*F*1*-Score*	*Precision*	*Recall*	*F*1*-Score*	*Precision*	*Recall*	*F*1*-Score*	*Precision*	*Recall*	*F*1*-Score*
HEALTHY	0.74	0.85	0.79	0.87	1.0	0.93	0.8	1.0	0.89	0.7	0.95	0.81
PHYVV	0.79	0.55	0.65	0.78	0.7	0.74	0.72	0.65	0.68	0.55	0.55	0.55
TMV	0.7	0.8	0.74	0.84	0.8	0.82	0.82	0.7	0.76	0.92	0.6	0.73
Macro avg	0.74	0.73	0.73	0.83	0.83	0.83	0.78	0.78	0.78	0.73	0.7	0.7

**Table 4 plants-10-01977-t004:** Recognition test accuracy of different fine-tuned models for different datasets (best results are highlighted with bold font).

Dataset	VGG-16 (avg/min/max)	Xception (avg/min/max)	Inception v3 (avg/min/max)	MobileNet v2 (avg/min/max)
RGB	93.3/91.6/93.6	93.3/91.8/93.4	91.8/91.6/93.8	93.3/91.8/93.8
NRBVI	93.3/91.8/95.9	96.6/95.2/97.9	93.8/91.6/95.9	91.8/83.3/93.8
NRBVI-Jet	91.6/89.7/91.9	95.0/91.8/96.6	89.7/87.7/90.0	87.76/83.3/89.8
NGBVI	**96.6/89.7/96.9**	**98.3/95.9/98.9**	**95.0/93.8/97.9**	**92.3/89.7/95.0**
NGBVI-Jet	95.0/81.6/95.2	93.3/81.2/94.7	89.7/85.7/91.8	91.8/89.7/93.4

**Table 5 plants-10-01977-t005:** Classification report of *precision*, *recall*, and *F*1*-score* of the models that use NRBVI-Jet dataset.

Class Name	VGG-16			Xception			Inception v3			MobileNet v2		
	*Precision*	*Recall*	*F*1*-Score*	*Precision*	*Recall*	*F*1*-Score*	*Precision*	*Recall*	*F*1*-Score*	*Precision*	*Recall*	*F*1*-Score*
HEALTHY	1.0	1.0	1.0	1.0	1.0	1.0	1.0	1.0	1.0	1.0	1.0	1.0
PHYVV	0.91	1.0	0.95	0.95	1.0	0.98	0.87	1.0	0.93	0.9	0.95	0.93
TMV	1.0	0.9	0.95	1.0	0.95	0.97	1.0	0.85	0.92	0.95	0.9	0.92
Macro avg	0.97	0.97	0.97	0.98	0.98	0.98	0.96	0.95	0.95	0.95	0.95	0.95

**Table 6 plants-10-01977-t006:** MobileNet v2 architecture description [33].

Input	Operator	Expansion Factor	Output Channels	Number of Repeated Layers	Stride
224^2^ × 3	Conv2d	-	32	1	2
112^2^ × 32	bottleneck	1	16	1	1
112^2^ × 16	bottleneck	6	24	2	2
56^2^ × 24	bottleneck	6	32	3	2
28^2^ × 32	bottleneck	6	64	4	2
14^2^ × 64	bottleneck	6	96	3	1
14^2^ × 96	bottleneck	6	160	3	2
7^2^ × 160	bottleneck	6	320	1	1
7^2^ × 320	bottleneck	-	1280	1	1
7^2^ × 1280	Avgpool 7 × 7	-	-	1	-
1 × 1 × 1280	Conv2d 1 × 1	-	-	k	-

## Data Availability

The datasets generated during and/or analyzed during the current study are available from https://doi.org/10.5281/zenodo.5500727 (accessed on 19 September 2021).

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
