# Peer review of "Analysis of New RGB Vegetation Indices for PHYVV and TMV Identification in Jalapeño Pepper (Capsicum annuum) Leaves Using CNNs-Based Model"

_plants, 2021, doi:10.3390/plants10101977_

Round 1
Reviewer 1 Report
This paper presents a model of using deep learning to distinguish side effects or other blemishes on plants. Here are concerns:
- We can find several deep learning models used in plant effects detection. What is new in your case? What is your model introducing to the field? Make better presentation and discus potential advances of your idea.
- Related ideas to extend: A survey of deep convolutional neural networks applied for prediction of plant leaf diseases, Self-attention Negative Feedback Network for Real-time Image Super-Resolution.
- Did you test other pooling and convolutions in your research? Can you justify proposed model composition?
- How was your model trained? Which algorithm was used? Make presentation and explain.
- Model needs comparisons to other like yolo, inception, coco, etc.
- Model needs verification on other data to confirm efficiency.
- What are limitations of your model? Is there any image resolution ok?
Author Response
Thank you very much for your comments.

Reviewer 2 Report
This paper tries to address PHYVV and TMV identification in Jalapeno pepper leaves by using CNN models. Extensive experiments show the effectiveness of the presented method. Before any possible publication of this paper, the authors are suggested to kindly address the following comments.
1, To facilitate understanding, the authors are suggested to re-organize the sections. Generally, results often show after the methodology description.
2, Generally, the generalization ability of methods is very important. I am wondering whether the adaptation of new TGB vegetation indices may help to improve the domain-invariant performance (e.g., Learning deep semantic segmentation network under multiple weakly-supervised constraints for cross-domain remote sensing image semantic segmentation. ISPRS JPRS; Unsupervised domain adaptation using generative adversarial networks for semantic segmentation of aerial images. Remote Sensing). Please give some comments on this.
3, The current version contains several spelling and grammar errors. The authors are suggested to carefully lift the presentation quality.
Author Response
Thank you very much for your comments.

Round 2
Reviewer 1 Report
With much regret I must confirm that none of my concerns is solved. There is no explained novelty in your applied ai model. There are no tests to compare to other models and also no tests on other data.
Author Response
COVER LETTER
COMMENT 1.- With much regret I must confirm that none of my concerns is solved. There is no explained novelty in your applied ai model. There are no tests to compare to other models and also no tests on other data.
Thank you for your comments. We attended the comments and explained details about the answer.
First, we described the most related state-of-art approaches for agricultural pest identification and classification that use CNNs (you can find this at lines 85 – 134 in the manuscript), and we explained the novelty of the proposed study. (this answers COMMENTS 1 and 2 of reviewer 1 in round 1)
In this study, we are not proposing a new AI deep learning techniques (model), instead the novelty of this project is the development of a new predictive analytics methodology to identify Tobacco Mosaic Virus (TMV) and Pepper Huasteco Yellow Vein Virus (PHYVV) visual symptoms for Jalapeño pepper (Capsicum annuum) leaves by means of new RGB vegetation indices as pre-processing algorithms and pre-trained deep learning models based on VGG-16, Xception, Inception v3, and MobileNet v2 architectures. The objective of this methodology is to find the most accurate combination of RGB vegetation index and deep learning classification model to identify PHYVV and TMV in Jalapeño pepper plants by using a small training dataset. Furthermore, it presents a measurement of how identification accuracy can be improved by preprocessing images with new RGB vegetation indices and its usefulness for future research.
Second, about “compare to other models”, we increased the experiments by adding new deep learning architectures, Inception v3 and MobileNet v2 as the reviewer 1 suggested in round 1 at COMMENT 5.
Please see the computational results in Tables 2 – 4 and Figures 2 -3. Furthermore, the description of Inception v3 and MobileNet v2 is at lines 441 – 468, and also in Figure 10 and Table 5.
We used Inception because it has demonstrated to attain good performance on complex computer vision datasets like the ImageNet dataset. For the case of MobileNet, YOLO and COCO are more utilized for real-time object detection, so we decided to use a light weighted CNN architecture. But we are interested to explore YOLO for object detection on future investigations. (this answers COMMENT 5 of reviewer 1 in round 1)
About the training process (this answers COMMENT 4 of reviewer 1 in round 1), we explained in detail, how we build and train the different CNNs-based models. Please see the Parameter setting of CNN architecture models in Table 1.
The models were trained using the Stochastic Gradient Descent (SGD) as an optimizer with 0.9 momentum, and the learning rate was varied between 1e-3 and 1e-4. The learning rate defines the learning progress of the proposed model and updates the weight parameters to reduce the loss function of the network. The maximum number of epochs was set to 100 and batch size of 16 were used in this experiment.
Furthermore, we explained how we split the data. Each dataset has 100 images for each class, for a total of 300 images. Each dataset was divided into 80% for training (240 images) data and 20% for testing (60 images, 20 images per class) data. During the training phase, 20% of validation data was randomly selected from the 240 images. This validation data is used to manually adjust hyperparameters, which are essentially the settings that cannot be automatically learned during training. These include the learning rate, and the batch size, among others.
And finally, about test our methodology with other data, we are planning to use other datasets (however, those evaluations were beyond the scope of this article), but for the proposed datasets. We did an exhaustive statistical analysis of the different models that we build. For example, for each dataset and CNN architecture, a total of 10 trials were carried out, for each model from scratch and 10 trials from transfer learning, with a random selection of 20 % for validation data, i.e., we build 40 models for each dataset and 200 models in each experimental configuration (models from scratch and from transfer learning) for a total of 400 models.
As we discussed in previous sections, the objective of this predictive analytics methodology is to study the usefulness of new RGB vegetation indices to improve classification accuracy and the most accurate combination of vegetation index and deep learning classification model to identify PHYVV and TMV in Jalapeño pepper plants by using a small training dataset. We highlighted that previous investigations have applied deep learning algorithms on raw RGB data without considering the previous image improvements of new vegetation indices.

Reviewer 2 Report
The authors have addressed my concerns. I think the current version can be considered to be published.
Author Response
COVER LETTER
We want to thank reviewer 2 for the precise comments on this article, his comments were useful to strengthen this work and gave us ideas for future research.
Note: the manuscript was edited by a native English speaker to fix grammar issues.

Round 3
Reviewer 1 Report
Thank you for explanations. I still miss tests and comparisons with detailed explanation of training and model development. I evaluate this paper as a novel idea for machine learning model, and as Authors say this is oriented on potential application without any new proposal. I think this paper would better fit some conference as nice discussion from some use of ready set methods.
